# Intervalley scattering by acoustic phonons in two-dimensional MoS$_2$ revealed by double-resonance Raman spectroscopy

Bruno R. Carvalho[1,*], Yuanxi Wang[2,*], Sandro Mignuzzi[3,4], Debdulal Roy[3,4], Mauricio Terrones[2,5,6], Cristiano Fantini[1], Vincent H. Crespi[2], Leandro M. Malard[1] & Marcos A. Pimenta[1]

Double-resonance Raman scattering is a sensitive probe to study the electron-phonon scattering pathways in crystals. For semiconducting two-dimensional transition-metal dichalcogenides, the double-resonance Raman process involves different valleys and phonons in the Brillouin zone, and it has not yet been fully understood. Here we present a multiple energy excitation Raman study in conjunction with density functional theory calculations that unveil the double-resonance Raman scattering process in monolayer and bulk MoS$_2$. Results show that the frequency of some Raman features shifts when changing the excitation energy, and first-principle simulations confirm that such bands arise from distinct acoustic phonons, connecting different valley states. The double-resonance Raman process is affected by the indirect-to-direct bandgap transition, and a comparison of results in monolayer and bulk allows the assignment of each Raman feature near the **M** or **K** points of the Brillouin zone. Our work highlights the underlying physics of intervalley scattering of electrons by acoustic phonons, which is essential for valley depolarization in MoS$_2$.

[1] Departamento de Física, Universidade Federal de Minas Gerais, Belo Horizonte, Minas Gerais 30123-970, Brazil. [2] Department of Physics and Center for 2-Dimensional and Layered Materials, The Pennsylvania State University, University Park, State College, Pennsylvania 16802, USA. [3] National Physical Laboratory, Hampton Road, Teddington TW11 0LW, UK. [4] Department of Physics, King's College London, Strand, London WC2R 2LS, UK. [5] Department of Chemistry, The Pennsylvania State University, University Park, State College, Pennsylvania 16802, USA. [6] Department of Materials Science and Engineering, The Pennsylvania State University, University Park, State College, Pennsylvania 16802, USA. * These authors contributed equally to this work. Correspondence and requests for materials should be addressed to B.R.C. (email: brunorc@fisica.ufmg.br) or to L.M.M. (email: lmalard@fisica.ufmg.br) or to M.A.P. (email: mpimenta@fisica.ufmg.br).

The second-order Raman spectrum of $MoS_2$ and other semiconducting transition-metal dichalcogenides (TMDs) hosts a rich variety of features, which are strongly dependent on the number of layers and the excitation laser energy[1–10]. In the published works on two-dimensional (2D) $MoS_2$, the second-order Raman bands were probed using only few laser excitation lines, and results were interpreted on the basis of phonon dispersion relations calculated from empirical force field models[4]. However, the use of multiple excitation energies is pivotal to unveiling the rich physical phenomena underlying the complex second-order Raman spectrum and the intervalley scattering processes in $MoS_2$.

Double-resonance Raman (DRR) is a special kind of second-order process that involves the resonant scattering of excited electrons by phonons, and can be used to study electrons, phonons and their interplay[11]. By varying the incoming photon energy, the DRR condition selects different electronic states and different pairs of phonons with opposite finite momenta within the interior of the Brillouin zone (BZ)[12–18]. For graphene, the most important DRR features, the D and 2D bands, provide rich physical information about the sample[12,19].

In this work, we unravel the origin of the DRR processes occurring in $MoS_2$ by investigating both experimentally and theoretically the Raman spectra as a function of the laser excitation. We measure the second-order bands using more than twenty different laser excitations ranging from 1.85 to 2.18 eV that densely cover the range of the A and B excitonic levels. Notably, we observe that the spectral positions of some specific second-order peaks depend on the laser excitation energy, which is characteristic of DRR processes. Results are explained by accurate first-principles calculations, which are shown to be crucial for identifying the different contributions to the DRR process. We show that some specific Raman processes are related to phonons near (but not at) the $\mathbf{M}$ and $\mathbf{K}$ points of the BZ. Moreover, the different contributions from the $\mathbf{M}$ phonons in monolayer and bulk reflect the crossover of the indirect-to-direct bandgap transition in the monolayer regime of $MoS_2$. Our work provides a fundamental explanation of the resonant behaviour of the second-order Raman processes in $MoS_2$, involving different electronic valleys, and may also be applicable to the higher-order Raman spectra and to other semiconducting TMDs.

## Results

**Experimental analysis.** Figure 1a,b show the Raman spectra of monolayer (1L) and bulk $MoS_2$ in the 350–500 cm$^{-1}$ spectral range for three different laser energies (1.94, 2.04 and 2.11 eV). The two peaks around 388 and 407 cm$^{-1}$ are associated, respectively, with the first-order in-plane and out-of-plane Raman bands, with $E'$ ($E_{2g}^1$ for bulk) and $A_1'$ ($A_{1g}$ for bulk) symmetries[10,20]. All other features are contributions from different second-order processes.

A first-order Raman band can be fitted with a Lorentzian curve, since it arises from a single phonon at the center of the BZ. On the other hand, a second-order Raman band is given by the convolution of multiple two-phonon processes across the whole BZ and, therefore, cannot be fitted by a sum of Lorentzian curves. The determination of the lineshape requires a complete theoretical description of Raman intensities, including electron-photon and electron-phonon coupling matrix elements. Nevertheless, the fitting of second-order bands in $MoS_2$ and other TMD compounds by a sum of Lorentzian curves has been widely used in the literature[3,4,7,10,21], because it provides a means to associate a feature in the spectrum with a specific phonon at a high-symmetry point within the BZ. Various works in the literature have fitted the second-order bands of $MoS_2$ with different numbers of Lorentzians, and given the resulting peaks

different assignments (or when unassigned, different names)[3,4,21]. This fitting procedure can yield different peak numbers, positions, widths and intensities[4,10,12,21]. Here we also use this procedure, but we stress that it is only intended to provide a reliable estimate of the spectral position of the different contributions to the second-order Raman bands. The central goal of this work is to achieve a quantitative comparison between the measured and the calculated spectra within the entire spectral range considered here while making use of a dense sampling in terms of laser energies. We impose constant values for the full width at half maximum as a constraint in our Raman analysis, leaving intensity and position unconstrained. This procedure was adopted to decrease the number of fitting parameters, since the full width at half maximum is not expected to depend significantly on the laser energy within this narrow range of energy (1.85–2.18 eV). The number of Lorentzian peaks is increased until a pre-defined convergence threshold is reached, and the results of this procedure are confirmed by first-principles calculations, as described below.

In Fig. 1a one observes a band at ca. 420 cm$^{-1}$ that we call $p_1$, and a broad and asymmetric band centered around 460 cm$^{-1}$, which is commonly called the 2LA band in the literature. In this work, this broad band was fitted by four Lorentzian peaks: a first peak around 440 cm$^{-1}$ and the other three denoted by $p_2$, $p_3$ and $p_4$, as shown in Fig. 1a. The same number of peaks was used to fit the spectra of bulk $MoS_2$, since more peaks were not found to be necessary to the fitting (in terms of convergence). This is possibly because the phonon branch splittings induced by interlayer interactions are not larger than $\sim 3$ cm$^{-1}$ in $MoS_2$ (ref. 22).

Figure 1c,d show the multiple excitation Raman map for 1L and bulk $MoS_2$ obtained using more than twenty laser lines with energies from 1.85 to 2.18 eV. The horizontal scale represents the Raman shift, while the vertical scale is the laser excitation energy. We can observe the resonances of all Raman bands across the A ($\sim 1.89$ eV) and B ($\sim 2.06$ eV) excitonic transitions, which are marked by horizontal dashed lines. An important result is the dispersion of some Raman features as the laser energy changes, as clearly revealed by the dashed lines tracking Raman peak positions (Fig. 1a–d).

Each second-order band in the range 400–480 cm$^{-1}$, measured with different laser excitation energies, was fitted according to the same procedure used in Fig. 1a,b. The positions of $p_1$, $p_2$, $p_3$ and $p_4$ as a function of the laser excitation energy are plotted in Fig. 1e,f for 1L and bulk $MoS_2$, respectively. Notably, the position of the $p_2$ peak is unchanged for all laser energies, whereas $p_1$, $p_3$ and $p_4$ red-shift as the laser excitation energy increases.

**Theoretical model.** To explain the experimental results, we calculated the second-order Raman spectra for 1L and bulk $MoS_2$ within the single-particle picture, using the electronic structure and phonon dispersion obtained from density functional theory (DFT) (see Supplementary Note 1 and Supplementary Fig. 1) for different laser excitation energies, as described in Theoretical methods.

Figure 2a shows a schematic representation of a DRR process in $MoS_2$, where the low-energy electronic structure is represented by parabolic bands at the BZ edges. The DRR process begins with an incoming photon creating an electron-hole pair of wave vector $\mathbf{k}$ near the $\mathbf{K}$ valley. The electron is then inelastically scattered by the emission of a phonon with branch index $\mu$, wave vector $-\mathbf{q}$, and energy $\hbar\omega_{-\mathbf{q}}^{\mu}$ to the $\mathbf{K}'$ valley. After that, the electron is inelastically scattered back to the $\mathbf{K}$ valley by the emission of a second phonon with branch index $\nu$, wave vector $\mathbf{q}$ and energy $\hbar\omega_{\mathbf{q}}^{\nu}$, where the electron-hole pair recombines emitting a photon with energy $E_L - \hbar\omega_{-\mathbf{q}}^{\mu} - \hbar\omega_{\mathbf{q}}^{\nu}$ (Stokes scattering). At most

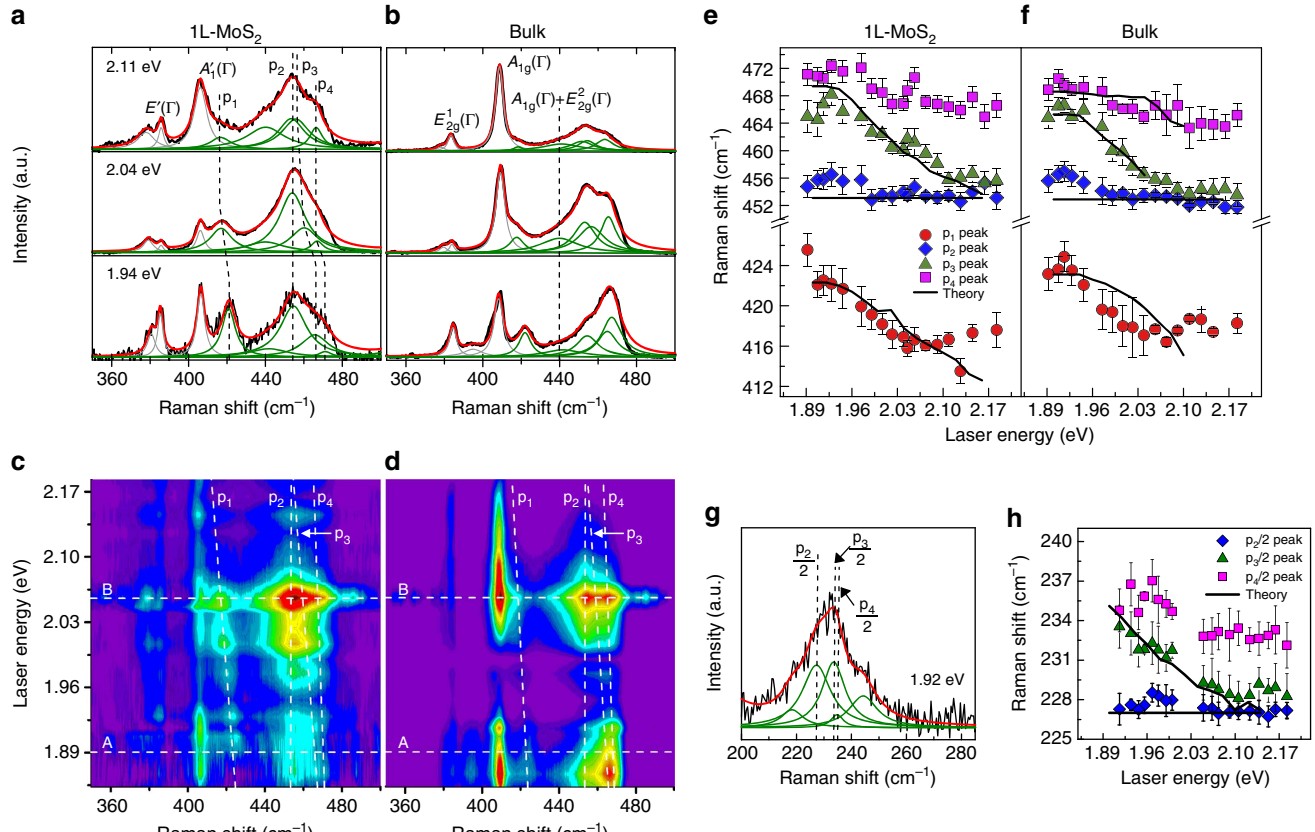

**Figure 1 | Resonance Raman results on 1L and bulk MoS$_2$.** (**a,b**) Raman spectra of 1L and bulk MoS$_2$ measured with three laser excitation energies: 1.94, 2.04 and 2.11 eV. The spectra are fit to a sum of Lorentzians (grain and green curves). The second-order bands studied here are denoted p$_1$, p$_2$, p$_3$ and p$_4$. (**c,d**) Resonant Raman maps of 1L and bulk MoS$_2$ acquired using more than twenty different laser lines, showing the enhancement of the Raman bands across the A and B excitons (horizontal dashed lines). (**e,f**) Laser energy dependence of the experimental values of the positions of the p$_1$, p$_2$, p$_3$ and p$_4$ peaks (symbols) of 1L and bulk MoS$_2$, and the calculated dispersion of these peaks using equation (1) (solid curves). (**g**) Raman spectrum of a defective 1L MoS$_2$ sample, showing the disorder-induced bands associated with acoustic phonons near the edges of the Brillouin zone. Defects were created through bombardment with Mn$^+$ (see Experimental Methods section for details). The band was fit to a sum of Lorentzians with frequencies half the frequencies of p$_2$, p$_3$ and p$_4$. (**h**) Laser energy dependence of the experimental values of p$_2$/2, p$_3$/2 and p$_4$/2 (symbols) showed in **g**, and the calculated dispersion of these disorder-induced peaks (solid curves) assuming one-phonon-defect DRR scattering. The absence of the theoretical curve of p$_4$ peak in **e,h** is due to its weak intensity not observable in the calculated Raman spectra (see Fig. 3 and, main text for details). The error bars in **e,f,h** represent the standard error from the fitting process.

two of these steps can be simultaneously resonant for a DRR process[9,11–13].

The intensity of a second-order Raman process is given by following expression:

$$I_{ee}^{pp}(E_L)=\left|\sum_{\mathbf{k},\mathbf{q},\mu,\nu}\frac{M_f\mathcal{M}_{cb}\mathcal{M}_{ba}M_0}{\left(E_L-E_{\mathbf{k}}^c+E_{\mathbf{k}}^v-\hbar\omega_{-\mathbf{q}}^\mu-\hbar\omega_{\mathbf{q}}^\nu-i\frac{\gamma}{2}\right)\left(E_L-E_{\mathbf{k}+\mathbf{q}}^c+E_{\mathbf{k}}^v-\hbar\omega_{-\mathbf{q}}^\mu-i\frac{\gamma}{2}\right)\left(E_L-E_{\mathbf{k}}^c+E_{\mathbf{k}}^v-i\frac{\gamma}{2}\right)}\right|^2,$$

(1)

where $M_0$, $M_f$ are the matrix elements of the exciton–photon interactions for the incoming and outgoing photons, $\mathcal{M}_{cb}$, $\mathcal{M}_{ba}$ represent the exciton–phonon interactions[23], $\mathbf{q}$ and $\mathbf{k}$ are the wave vectors of the phonon and the electron, respectively, $E_{\mathbf{k}+\mathbf{q}}^c$ and $E_{\mathbf{k}}^c$ are the energies of the intermediate states, and $\mu$ and $\nu$ denote the phonon branches involved in the process. The damping constant $\gamma$ is related to the finite lifetime of the intermediate states, and $\hbar\omega_{\pm\mathbf{q}}^{\mu,\nu}$ is the corresponding phonon energy. In the present study, we only focus on two-electron processes (ee) involving transverse acoustic (TA) and longitudinal acoustic (LA) phonons; processes involving hole scattering (hh, eh and he)[24] were not taken into account for clarity and will be elaborated in future studies. The matrix elements are taken to be constants in the calculations.

Following the usual analysis for DRR in graphite[25], the calculated DRR intensities are dominated by contributions from phonon wave vectors near $\mathbf{q}\sim0$ and $\mathbf{q}\sim2\mathbf{k}$ (distance from the **K** point), as shown in Supplementary Fig. 1. As the laser energy increases, the peak corresponding to $\mathbf{q}\sim0$ is almost dispersionless and the peak corresponding to $\mathbf{q}\sim2\mathbf{k}$ is dispersive[25]. It has been reported for graphene[26–28] that the $\mathbf{q}\sim0$ contribution vanishes due to the symmetry selections rules and quantum interference. We show later that, in the case of MoS$_2$, this contribution, which is present in the DFT results, should vanish if excitonic effects are considered. For now, we only focus on the $\mathbf{q}\sim2\mathbf{k}$ contribution by tracking the most dispersive Raman peaks in the calculated DRR intensities, since the $\mathbf{q}\sim0$ contributions are almost dispersionless (other dispersionless features such as van Hove singularities can be identified separately from the phonon density of states (pDOS), as shown later).

Figure 2b,c show a contour map of the conduction electron energies for 1L and bulk MoS$_2$, where the conduction band minimum occurs at **K** for 1L MoS$_2$. Based on the calculated electronic structure, Fig. 2d shows three sets of electronic states that participate in the DRR process in 1L MoS$_2$, for laser excitation energies of 1.9, 2.0 and 2.1 eV, represented,

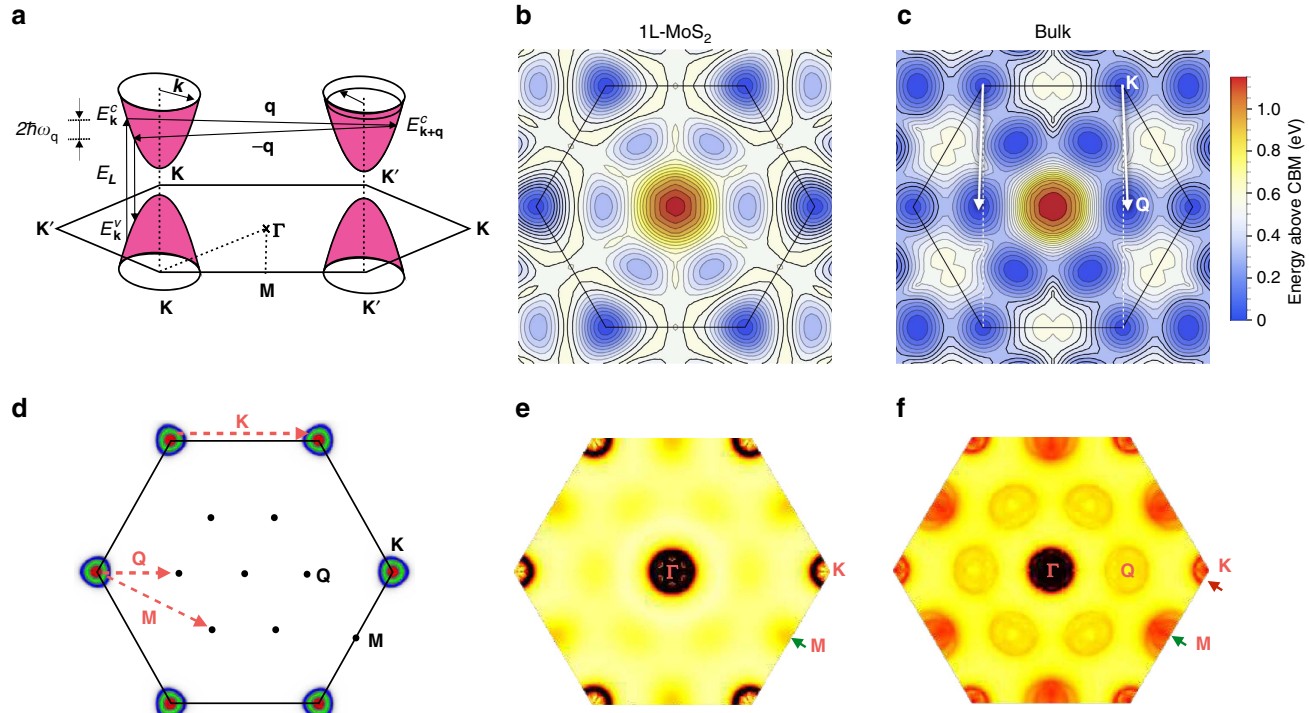

**Figure 2 | Double-resonance Raman model. (a)** Representation of a DRR process where excited electronic states are connected by two phonons with opposite momenta **q** and $-$ **q**. (**b,c**) Plot of the conduction bands for 1L and bulk MoS$_2$, respectively. (**d**) Locus of electronic states that participate in the DRR Raman process around **K** for laser energies of 1.9, 2.0 and 2.1 eV in red, green and blue. Possible phonon wave vectors that connect these states with each other and with other high-symmetry points (**M** and **Q**) are shown as red dashed arrows. (**e,f**) Density of states of phonons that satisfy the DRR conditions for 1L and bulk MoS$_2$, respectively. The DRR intensity in **e,f** is in arbitrary units justifying the use of no colour bars.

respectively, in red, green and blue. Similar to the case of graphene[19], when the laser excitation energy increases the locus of the on-resonance electronic states expands outward from **K**. Figure 2d also shows that phonons that connect the pockets around **K** and **K′** valleys have wave vectors near **K**, denoted $q_{\sim k}$ here (the symbol '$\sim$' means in the vicinity of the given high-symmetry point). On the other hand, pockets around the **K** and **Q** points are connected by phonons near both the **M** and **Q** points, with wave vectors $q_{\sim M}$ and $q_{\sim Q}$, where **Q** is another conduction band local minimum half-way between **Γ** and **K**, as shown in Fig. 2b–d.

Figure 2e shows the on-resonance density of states for acoustic phonons that connect two pockets in the electronic structure at 2.0 eV excitation for 1L MoS$_2$. A high density of $q_{\sim k}$ phonons appears around **K** and contributes strongly to the DRR scattering. A weaker density of phonons around **M** also appears (green arrow in Fig. 2e). They originate from scattering between the **K** and **Q** valleys (see Fig. 2d). For 1L MoS$_2$, the intervalley scattering between **K** and **Q** only contributes weakly to the DRR process since the two minima are misaligned by 0.2 eV for 1L MoS$_2$.

The above procedure is also applied to bulk MoS$_2$, where the conduction band local minimum at **Q** drops in energy, rendering the band gap indirect[29,30], as shown in Fig. 2c. Now, both $q_{\sim k}$ and $q_{\sim M}$ phonons in bulk MoS$_2$ contribute strongly to the on-resonance pDOS (red and green arrows in Fig. 2f), as well as the additional scattering channel between **K** and **Q** with phonon wave vector $q_{\sim Q}$ (see Fig. 2d).

Figure 3a–d show the calculated second-order Raman spectra for 1L and bulk MoS$_2$ for different laser energies, as given by equation (1), and using the phonon dispersion relations of the LA and TA phonons (Supplementary Fig. 1). In the range 405–420 cm$^{-1}$ (Fig. 3a,c), we observe one dispersive and one

non-dispersive feature, and in the range 450–470 cm$^{-1}$ (Fig. 3b,d), one non-dispersive and two dispersive features at higher frequency. A normal second-order Raman band is non-dispersive, whereas a dispersive behaviour indicates a DRR process. The positions of each feature in Fig. 3a–d as a function of the laser energy are also plotted in Fig. 1e,f, for 1L MoS$_2$ and bulk, respectively, and represented by the black lines. The comparison between the experimental and calculated results in Fig. 1e,f shows an excellent agreement, and will allow for the assignment of each second-order feature. As $q_{\sim k}$ expands away from **K**, the frequency of on-resonance LA phonons would decrease with increasing laser energy due to the negative curvature of the LA branch dispersion near **K**. Although p$_4$ is not clearly observable in the calculated DRR spectra for 1L MoS$_2$, a small contribution near $\sim$**M** can be seen in the BZ mapping of DRR intensity in Fig. 2e (green arrow). The underestimated intensity of this contribution from **M**, compared with the experimental results (where the contribution from **M** appears larger), might be due to strong electron-phonon coupling matrix elements between $\sim$**K** and $\sim$**Q** conduction valleys by an $q_{\sim M}$ phonon[27] (matrix elements are treated as constants in this study).

**Assignment of the second-order Raman features of MoS$_2$.** *The 2LA band.* The strongest feature in the second-order spectrum of MoS$_2$ is the broad and asymmetric band centered around 460 cm$^{-1}$. In previous Raman studies of bulk MoS$_2$ (ref. 1), it was initially ascribed to the overtone of the LA phonon at the **M** point in the BZ and assigned to the 2LA(**M**) band[1,2]. However, its asymmetric shape led some authors to suggest that it could have contributions from other second-order processes. Verble and Wieting[6] assigned the shoulder of the 2LA band at

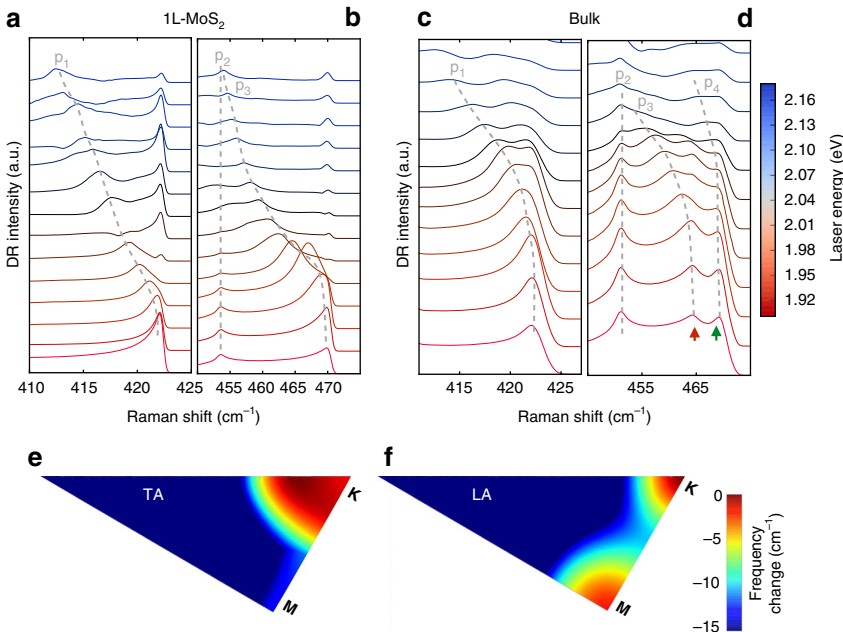

**Figure 3 | Calculated resonance Raman spectra.** Calculated second-order Raman for (**a,b**) 1L and (**c,d**) bulk MoS$_2$ for different laser energies, using equation (1). All spectra are vertically shifted for clarity. (**e,f**) Calculated phonon frequencies of the TA and LA branches with respect to the frequency maxima of each branch, showing that the LA branch disperses more rapidly away from **K**.

466 cm$^{-1}$ to the Raman-inactive mode with $A''_{2u}$ symmetry (or $A_{2u}$ mode for bulk or even layers) at the **Γ** point. Gołasa *et al.*[4] suggested this band was composed by a combination of the optical $E_{1g}$ phonon with the TA phonon at the **M** point. Livneh and Spanier[21] proposed different contributions for the 2LA band in bulk MoS$_2$, such as the pDOS, overtones of the **M** edge phonons from the acoustic (LA) and quasi-acoustic (LA$'$) branches, which are very close in energy, and a weak contribution of the LA(**K**) phonons.

In this work, we assign the different contributions to the 2LA band by comparing the experimental and calculated multiple excitation Raman spectra of both 1L and bulk MoS$_2$. In our analysis, the 2LA band was fitted by four peaks, a peak around 440 cm$^{-1}$ and the peaks p$_2$, p$_3$ and p$_4$, as shown in Fig. 1a,b. The non-dispersive behaviour of p$_2$ (see Fig. 1e,f) shows that it corresponds to a normal second-order Raman process. On the other hand, p$_3$ and p$_4$ exhibit a dispersive behaviour, a signature of a DRR process (or higher-order resonant process) due to photons with different energies selecting electrons and phonons with different wave vectors in the BZ. Our calculations show that p$_2$ comes from the van Hove singularity (vHs) in the pDOS from a saddle point between **K** and **M**. The frequency of p$_2$ matches with twice the calculated frequency of the vHs. Our assignment for p$_2$ agrees with the conclusion reached by Livneh and Spanier[21] (the L$_2$ peak in their work). The 440 cm$^{-1}$ peak was recently ascribed to the combination mode $A_{1g} + E^2_{2g}$ (ref. 21). However, it can also be due to the asymmetric shape of p$_2$, since the vHs is not necessarily symmetric.

We next discuss the origin of the p$_3$ and p$_4$ peaks. Previous works assigned them to 2LA(**M**) and/or 2LA(**K**) processes, but the near-degeneracy of the LA phonon at **M** and **K** prevented a clear distinction between these two contributions[1–5,31]. A comparison of our results for 1L and bulk MoS$_2$ allows for an unambiguous assignment. As shown in Fig. 2f, the DRR process in bulk MoS$_2$ can support scattering by both $\mathbf{q}_{\sim M}$ and $\mathbf{q}_{\sim K}$ phonons, whereas DRR in 1L MoS$_2$ (Fig. 2e) is very weak

for $\mathbf{q}_{\sim M}$ phonons. The different DRR contributions from $\sim$**M** phonons in 1L and bulk MoS$_2$ allow us to distinguish the contributions of DRR $\sim$**K** and $\sim$**M** phonons to the second-order Raman band, as will be shown below. The calculated results in Fig. 3b,d show that p$_4$ clearly appears in the spectra of bulk MoS$_2$, but is very weak for 1L MoS$_2$. Notice that different from the calculated results, the experimental intensity of p$_4$ for 1L MoS$_2$ is weak but not negligible, this is possibly ascribed to the effect of a strong electron-phonon coupling of p$_4$. However, the most relevant result is that p$_4$ is more intense than p$_2$ and p$_3$ for 2L, 3L and bulk MoS$_2$ (see Supplementary Note 2 and Supplementary Fig. 2), and weaker for 1L MoS$_2$.

We, therefore, conclude that the p$_4$ peak is related to the scattering of the excited electron between the **K** and the **Q** valleys by two LA phonons near $\sim$**M** (see Fig. 2d). This leaves p$_3$ to be assigned to the scattering process between **K** and **K$'$** by two LA phonons in the vicinity of the **K** point. Thus, a more precise assignment for p$_3$ and p$_4$ would be 2LA($\sim$**K**) and 2LA($\sim$**M**), respectively.

Figure 1e,f compare the experimental positions of p$_3$ and p$_4$ with their calculated dispersion. The excellent agreement between experiment and theory further confirms the assignment of p$_3$ and p$_4$ as 2LA($\sim$**K**) and 2LA($\sim$**M**). As the laser energy increases, p$_3$ and p$_4$ disperse at rates of $-49$ and $-21$ cm$^{-1}$ eV$^{-1}$, respectively, and this result reflects the different slopes of the LA phonon dispersion near **K** and **M** of MoS$_2$. We thus conclude that the broad and asymmetric band centered around 460 cm$^{-1}$ can be explained by contributions from LA phonons, instead of optical phonons[4,6]. It has contributions from LA phonons near the saddle point between **K** and **M** (pDOS singularities) and from LA phonons near $\sim$**K** and $\sim$**M**, which are enhanced in the spectra by the intervalley DRR process.

*The p$_1$ (or b) peak.* In a resonance Raman study of bulk MoS$_2$ at 7 K, Sekine *et al.*[32] observed a band around 430 cm$^{-1}$ with a dispersive behaviour near the A and B excitonic resonances, and called it the *b* band (see Supplementary Note 3 and

Supplementary Fig. 3). This band was ascribed to a two-phonon process involving a quasi-acoustic interlayer breathing mode and a transverse optical phonon $(E_{1u}^2)$ both along the *c*-axis[32]. However, this band was also observed in more recent Raman studies of monolayer MoS$_2$ (ref. 21), thus ruling out the assignment suggested by Sekine *et al.*[32]. In a detailed multi-phonon study of bulk MoS$_2$, Livneh and Spanier suggested that the *b* band could involve combinations of the LA (or LA′) and TA (or TA′) phonons at the **K** point[21].

In Fig. 1e,f we compare the experimental values of the p$_1$ positions (red circles) with the dispersion of the calculated peak (black curve) considering an intervalley DRR process involving one LA phonon and one TA phonon, in both time orders, in the vicinity of **K** (see Fig. 3a,c). The combined energy of the relevant LA + TA phonons near ∼**K** red-shifts as the incident laser energy increases, dispersing at a rate of $-43\,cm^{-1}\,eV^{-1}$. The agreement between the experimental and calculated results in Fig. 1e,f allows for the assignment of p$_1$ to LA($\sim$**K**) + TA($\sim$**K**).

The larger dispersion of p$_3$ = 2LA($\sim$**K**) than p$_1$ = LA($\sim$**K**) + TA($\sim$**K**) is consistent with TA phonons dispersing more slowly near **K** than do LA phonons. This can be seen in Fig. 3e,f, which show the phonon dispersion for the TA and LA branches in the irreducible BZ, where the frequencies are represented by a colour map. The LA phonon frequency rapidly decreases away from **K**, whereas the TA phonon frequency varies slowly near **K**.

*The 2LA($\sim$**Q**) band.* As represented in Fig. 2d, there are two different scattering processes between the **K** and **Q** electronic valleys, involving phonons with both wave vectors $\mathbf{q}_{\sim\mathbf{M}}$ and $\mathbf{q}_{\sim\mathbf{Q}}$. Indeed, the theoretical DRR maps in Fig. 2f predict the existence of a 2LA($\sim$**Q**) band for bulk MoS$_2$ (see the red circle labelled ∼**Q** in Fig. 2f) at a frequency of 370 cm$^{-1}$. This band was not detected in our room-temperature spectra, presumably because the electron-phonon coupling matrix elements for this process are almost vanishing[27]. However, it was possibly observed in previous low-temperature studies in bulk MoS$_2$ (refs 21,32). Sekine *et al.*[32] observed a weak feature around 380 cm$^{-1}$ in the spectra at 7 K, and ascribed it to the zone-center Raman-inactive $E_{2u}^1$ phonon. Livneh and Spanier reported the existence of a band at 386 cm$^{-1}$, observed at 95 K, which could not be explained by their multi-phonon analysis[21]. A multiple excitation Raman study at low temperatures will be necessary to confirm the existence of the double-resonance 2LA($\sim$**Q**) band, and its possible dispersive behaviour.

**Disorder-induced DRR bands**. As a further verification of our proposed DRR conclusions involving the acoustic phonons, the defect-induced resonant Raman spectra in 1L MoS$_2$ were measured. Figure 1g shows the spectrum in the range 200–285 cm$^{-1}$, where we can observe defect-induced Raman features[33] for a excitation energy of 1.92 eV (see also Supplementary Note 4 and Supplementary Fig. 4). This process is similar to the disorder-induced D band in graphene[19], which also comes from a DRR process involving just one phonon, where momentum conservation is provided by elastic scattering of the excited electron by a defect[9,12,19]. We have fitted the defect-induced Raman band with four Lorentzian peaks, similar to the procedure used for the 2LA second-order band in 1L pristine MoS$_2$. The frequencies of the three peaks at 227, 234 and 235 cm$^{-1}$ correspond, respectively, to one-half the frequencies of the p$_2$, p$_3$ and p$_4$ peaks as marked by the vertical dashed lines in Fig. 1g. Figure 1h shows the frequencies of these three Raman peaks as a function of the laser excitation energy, and the calculated dispersion of the DRR features considering now scattering by a phonon and a defect. The excellent agreement with the theory and experiment further supports the interpretation of the second-order DRR features, and highlights the intervalley elastic scattering by defects.

**Excitonic effects**. We now discuss the validity of the single-particle picture in the presence of excitonic effects. Single-particle DFT calculations have been extensively used in studying DRR processes in graphene, where excitonic effects are negligible at low excitation energies[34]. The strong excitonic effects in 1L MoS$_2$ call for further interrogation of assignments based on single-particle DFT results. Here based on a simple two-band model, we show that the use of single-particle band structures (see Supplementary Note 5 and Supplementary Fig. 5a,c,e) and exciton band structures (see Supplementary Note 5 and Supplementary Fig. 5b,d,f) would yield similar laser-energy dependence of the Raman frequencies, thereby justifying the use of DFT eigenvalues in equation (1).

In the single-particle case, we model the low-energy band structure as parabolic bands $E_{\pm} = \pm(E_g/2 + \mathbf{k}^2/2m^{\star})$, where for convenience we assume that $m_e = m_h \equiv m^{\star}$ is the mass of the electron and hole and that the conduction band has been rigidly shifted to bring the single-particle band-gap to match with the optical gap $E_g$. The actual electron-hole mass ratio calculated from different first-principles methods ranges from 0.8 to 1.8 (refs 35–38).

From the single-particle bands, excitons can be approximately represented by pairing electrons and holes with matching group velocities at $\mathbf{k}_e = -\mathbf{k}_h \equiv \mathbf{k}$ (where **k** is the wave vector measured from **K**); such an exciton has a center-of-mass momentum of $\mathbf{Q} = 2\mathbf{k}$ and a total energy of $E_g + 2(\mathbf{k}^2/2m^{\star}) = E_g + \mathbf{Q}^2/2M$. Thus, we find $M = 2m^{\star}$, that is, the excitonic dispersion has half the band curvature of the single-particle bands. This is consistent with the semi-classical interpretation that the mass of an exciton is the sum of its constituents' masses. Momentum-resolved electron energy-loss spectroscopy measurements[39] are consistent with the exciton mass being larger than the constituent hole and electron masses obtained from first-principles calculations: 0.9–1.4 times their sum, depending on the exchange-correlation functional and pseudopotential used (note that the discussion below still applies, semi-quantitatively, for a reasonable range of exciton masses). Finally, we assume that the excitonic dispersion near **Q** = 0 and **Q** = **K** can be described by parabolic bands with the same band minimum and same curvature, since they both describe electron-hole pairs near the two degenerate valleys. That is, the energy difference of the two minima calculated by solving the tight-binding-based Bethe-Salpeter equation, 0.015 eV, is neglected here[40].

We now discuss intervalley scattering between single-particle states and between excitonic states within the above assumptions, as depicted in Supplementary Fig. 5c,d. When the laser energy is $E_L = E_g + 2\mathbf{k}^2/2m^{\star}$, the on-resonance phonons in the single-particle bands are 2**k** away from **K**, as previously described. When the excitonic dispersion replaces the single-particle energies in equation (1), the resonance condition will be satisfied only if the incoming photon matches the optical band gap, that is, $E_L = E_g$, where excitons are scattered to **Q** = **K** and back. However, as the laser energy increases beyond the optical gap $E_L > E_g$ (but still within the exciton linewidth), a single and weaker resonance can still be achieved when the exciton is scattered to the other valley near **Q** = **K**, and then back. The same laser energy as the one given in the single-particle system can be recast in terms of the exciton mass $E_L = E_g + 2\mathbf{k}^2/2m^{\star} = E_g + (2\mathbf{k})^2/2M$. Hence the on-resonance phonons scattering excitons are also away from **K** by 2**k**, the same as the single-particle case. Therefore, the excitation-energy dependence of the Raman frequencies (but not the intensities) can be reasonably well described by a single-particle

electronic band structure, as both pictures give the so-called $q \sim 2k$ contribution of on-resonance phonons. We note that the $q \sim 0$ contribution, which appears in the single-particle picture, is lost in the excitonic picture.

The single-particle results may also carry over to the many-body picture due to the presence of trions. Unlike neutral excitons with vanishing momenta, trions of finite momenta can be created where the momentum comes from the extra charge carrier. Thus, the momentum of this many-body state again depends on the laser excitation energy, and a dispersive behaviour is expected.

Our conclusion that the LA($\mathbf{K}$) phonon dominates the intervalley scattering is consistent with previous reports of valley depolarization in TMDs[41–43]. From the decay of the valley polarization with increasing temperature, Zeng et al.[41] extracted the energy of the phonon mode dominating intervalley scattering to be $240 \, \text{cm}^{-1}$, consistent with the LA($\mathbf{K}$) phonon energy. However, this thermal destruction of valley polarization could also originate from the excitonic levels shifting as a function of temperature, thus detuning the laser excitation energy from the A exciton energy[42]. To rule out this possibility, Kioseoglou et al.[43] performed depolarization measurements as a function of increasing pump laser energy at $T = 5 \, \text{K}$, whereby the excess energy enables phonon-assisted intervalley scattering. These authors found that the destruction of the valley polarization occurs beyond an excess energy of 60 meV ($\approx 480 \, \text{cm}^{-1}$), which corresponds to two-LA($\mathbf{K}$) phonons, thus showing that intervalley scattering by LA phonons is responsible for depolarization[43].

## Discussion

To summarize, this work explains the origin of the DRR process in monolayer and bulk $MoS_2$, as involving different intervalley scattering processes. The resonant Raman spectra of the most intense second-order features in $MoS_2$ and the associated first-order disorder-induced bands, were measured using many laser excitation energies in the range of 1.85–2.18 eV, which covers the A and B excitonic levels. Experimental results are explained by DFT calculations of the DRR scattering in monolayer and bulk $MoS_2$. We demonstrate that the use of multiple excitation energies is crucial for understanding the physical phenomena underlying the second-order Raman spectrum of $MoS_2$.

We observe that the spectral position of some specific second-order peaks depends on the laser excitation energy, which is characteristic of DRR processes. By varying the incoming photon energy, the DRR condition selects different electronic states in the $\mathbf{K}$ and $\mathbf{Q}$ valleys, and different pairs of phonons with opposite finite momenta near (but not at) the $\mathbf{M}$ and $\mathbf{K}$ points of the BZ. Our results show that the DRR process reflects the indirect-to-direct bandgap transition from bulk to monolayer, and this effect allows the assignment of the Raman features to specific phonons near $\mathbf{M}$ or $\mathbf{K}$ (see also Supplementary Note 6 and Supplementary Tables 1–3).

Our study can also be extended to explain the second-order Raman spectra and the double-resonance process in other semiconducting TMDs, such as $MoSe_2$, $WS_2$, $WSe_2$. Moreover, the methodology in this work, based on multiple excitation Raman results and first-principle calculations, can also be used to explain the multiphonon spectra of semiconducting TMDs that exhibit a rich variety of high-frequency features, up to the fifth-order[4,21,44]. Finally, we show the second-order DRR spectra of $MoS_2$ originates in intervalley scattering by acoustic phonons, a mechanism which is also responsible for the destruction of valley polarization (that is, depolarization)[41,43]. Our work is thus relevant for the field of valleytronics of $MoS_2$, since the robustness of valley polarization depends sensitively on the absence of intervalley scattering[41,42,45].

## Methods

**Experimental methods.** The $MoS_2$ samples were obtained by mechanical exfoliation of natural 2H-$MoS_2$ crystals transferred onto Si substrates with a 298 nm thick $SiO_2$ coating. Defective monolayer $MoS_2$ was created through bombardment with $Mn^+$ in an ultrahigh vacuum time-of-flight secondary ion mass spectrometry (TOF-SIMS IV) instrument (ION-TOF GmbH, Muenster, Germany), equipped with a liquid metal ion gun at an angle of 45° to the surface normal and using an ion-beam kinetic energy of 25 keV, as described in ref. 33. The ion current and the exposure time were tuned to obtain an average inter-defect distance of 2.2 nm. Micro-Raman measurements were obtained on a DILOR XY triple-monochromator spectrometer equipped with a $N_2$-cooled charge-couple device detectors and with $1,800 \, \text{g} \, \text{mm}^{-1}$ diffraction gratings, giving spectral resolution better than $1 \, \text{cm}^{-1}$. The $MoS_2$ samples were excited using different laser sources (Ar/Kr, and dye laser with DCM special and rhodamine 6G) covering excitation energies from 1.85 to 2.18 eV. All measurements were conducted at room temperature in backscattering geometry where the laser propagation is perpendicular to the $MoS_2$ layer plane. A $100 \times$ objective provided a spot of $1 \, \mu\text{m}$ diameter. The laser power at the sample surface was kept below 1.0 mW to avoid sample heating. The Raman spectra were normalized by the intensity of the Si peak at each excitation energy and, the Si Raman peak at $521.6 \, \text{cm}^{-1}$ is used to calibrate the Raman shift.

**Theoretical methods.** DFT calculations were performed using the generalized gradient approximation parametrized by Perdew-Burke-Ernzerhof for the exchange correlation functional[46]. Electron–nucleus interactions are described within the projector augmented wave formalism[47,48]. All structural relaxation and band structure calculations were carried out by the Vienna Ab-initio Simulation Package[49,50], with energies converged at a plane wave expansion energy cutoff of 400 eV and forces converged at $0.004 \, \text{eV} \, \text{Å}^{-1}$. Interatomic force constants were calculated using the supercell method with the size of the supercell converged at $6 \times 6 \times 1$. Interlayer van der Waals interactions were included using the semi-empirical DFT-D2 method[51]. All conduction bands calculated from DFT have been rigidly shifted to match with the experimentally measured optical gaps for 1L and bulk $MoS_2$. Resonant Raman calculations were performed using phonon dispersions sampled on a $400 \times 400 \times 1$ $\mathbf{\Gamma}$-centered grid for 1L $MoS_2$ and a $200 \times 200 \times 50$ grid for bulk. A phenomenological damping constant (smearing) of $\gamma = 0.02 \, \text{eV}$ was used. Since the absolute Raman shifts are prone to slight (1%) over/under-estimates due to limitations of DFT, the calculated phonon frequencies were shifted by within 1% to allow better comparison with the respective experimental dispersions. This shift leaves the calculated dispersion in the Raman shift with laser energy (which reflects the influence of the phonon and electron band structures on the DRR process) unchanged.

**Data availability.** All relevant data are available from the authors upon request.

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

## Acknowledgements

We thank Profs Tony Heinz, James Hone and Changgu Lee for the pristine samples preparation and Profs Kin Fai Mak and Tsachi Livneh for helpful discussions. B.R.C., C.F., L.M.M. and M.A.P. acknowledge the financial support from the Brazilian agencies CNPq, CAPES, FAPEMIG and Brazilian Nanocarbon Institute of Science and Technology (INCT-Nanocarbono). Y.W., M.T. and V.H.C. acknowledge support from the U.S. Army Research Office MURI grant W911NF-11-1-0362. Y.W. and V.H.C. acknowledge support from the NSF Materials Innovation Platform program under DMR-1539916. M.T. acknowledges support from the National Science Foundation for the grants no. 2DAREEFRI-1433311. S.M. and D.R. acknowledge financial support from the Innovation Research and Development Programme of the National Measurement System, UK, Project No. 115948 and ChemBio program on Raman Metrology. S.M. conveys thanks to Dr Barry Brennan for help in sample preparation. Open access for this article was partly funded by Physics graduate program at the Universidade Federal de Minas Gerais, Penn-State University and King's College London.

## Author contributions

B.R.C., L.M.M. and M.A.P. conceived the idea and designed the experiments; B.R.C., C.F. and L.M.M. performed the resonance Raman experiments; B.R.C. analysed and interpreted the experimental data; Y.W. and V.H.C. performed the density functional theory calculation and theoretical analyses; S.M. and D.R. explored the preliminary concept and prepared the defective 1L $MoS_2$ sample; B.R.C., Y.W. and M.A.P. wrote the paper. The work was supervised by M.T., V.H.C. and M.A.P.; All the authors discussed the results and commented on the manuscript.

## Additional information

**Competing financial interests:** The authors declare no competing financial interests.

**Publisher's note**: 

