## [Peer Review File · Nature Communications]

Reviewers' Comments:

Reviewer #1 (Remarks to the Author)

In their manuscript entitled 'Intervalley scattering by acoustic phonons in two-dimensional MoS₂ revealed by double-resonance Raman spectroscopy' Carvalho et al. report on excitation-energy dependent double-resonant Raman scattering in a transition metal dichalcogenide monolayer and bulk crystal and discuss the physical origin of the observed Raman modes via comparison with DFT calculations. Furthermore, the model is then extended from the single-particle to the excitonic framework.

The paper is well written and intervalley scattering processes are of importance for future work in the field of valleytronics. However, I am not fully convinced of the relevance of the present work for the broad readership of Nature Communications as well as of the data analysis. My major concerns are the following:

- In order to fit the experimental spectra, a large number of symmetric Lorentzians is used. However, not every peak needed for the fit has a physical meaning, e.g., the peak at 440cm^{-1} . It could simply arise from the asymmetry of the p₂ peak. Why do the authors exclude asymmetric lineshapes in their fitting procedure?
- What are the different contributions to the error bars in Fig. 1(e),(f) and (h)?
- The authors claim that the p₄ feature, which should be absent in a monolayer, has an almost negligible intensity in the single-layer sample. However, in Fig. 1(a) at an excitation energy of 2.11 eV, the p₄ feature is of comparable intensity to the p₁, p₂, and p₃ peaks. Hence, the interpretation of the p₄ feature is not really sound.
- From the manuscript it doesn't become clear that for the measurements shown in Fig. 1(g) an ion bombarded sample is used. This should be mentioned more prominently. Additionally, the Raman spectrum in the same spectral range as in Fig. 1(g) of a pristine flake should be shown. How does the ion bombardment influence the high-energy p₁-p₄ peaks? This would be an important information in order to get more insight into the intervalley scattering processes in the presence of defects.
- The influence of the intervalley scattering effects described in the present work on a valley-polarisation of excitons should be discussed more extensively.

Reviewer #2 (Remarks to the Author)

The paper by Carvalho et al. is in my opinion a very good piece of science. It is amazing how the authors provided a very detailed description of acoustic phonon dynamics in 2D MoS₂. The paper reports an extremely careful resonance Raman experiments by using many laser energies, a feature that is really needed for carrying out such experiments. The theoretical model employed at the state of the art in condensed matter physics fully support the experimental data and allowed the authors to further understanding the intervalley scattering by acoustic phonons using double resonance Raman spectroscopy. The detailed description of the defect induced modes is also a great advance that this paper brings to the community, especially the elucidation of the intervalley scattering by acoustic phonons and its consequence to valleytronics in these kind of materials. I strongly recommend the publication of the paper in Nature Communications because the results presented here are sound and will broadly impact the field of 2D materials, a hot topic today in materials science.

Before the final acceptance, the authors should comment some points.

Page 3. The frequency dependence of some modes is attributed to DRR mechanism. Is the DRR the unique way of having frequency dependence on laser excitation? The authors should clarify better this point.

Page 4. Is there any physical reason that would justify the linewidth fixed as a constraint for the Raman analysis?

Could the authors comment on the kind of defect will activate the disorder induced modes in MoS₂? Is possible by using symmetry arguments predicts some edge dependence of disorder induced modes in MoS₂? How polarization depend measurements will help to further understand de intervalley scattering in MoS₂?

Reviewers' comments:

Reviewer #1 (Remarks to the Author):

In their manuscript entitled 'Intervalley scattering by acoustic phonons in two-dimensional MoS₂ revealed by double-resonance Raman spectroscopy' Carvalho et al. report on excitation-energy dependent double-resonant Raman scattering in a transition metal dichalcogenide monolayer and bulk crystal and discuss the physical origin of the observed Raman modes via comparison with DFT calculations. Furthermore, the model is then extended from the single-particle to the excitonic framework.

The paper is well written and intervalley scattering processes are of importance for future work in the field of valleytronics. However, I am not fully convinced of the relevance of the present work for the broad readership of Nature Communications as well as of the data analysis.

Answer: We thank the reviewer for the constructive comments and criticisms. We believe that our work is novel and relevant since we report the first systematic study unveiling the intervalley scattering mechanism by acoustic phonons in 2D MoS₂, which is essential for future valleytronics applications of these materials and other semiconducting TMD systems. It is also the first complete study fully describing the double-resonance Raman process in MoS₂, thus becoming a fundamental tool for characterizing different physical properties of 2D materials beyond graphene. For this reason, we believe that our work meets the criteria of broad readership required by Nature Communications. Below we address all the comments raised by the Reviewer.

My major concerns are the following:

In order to fit the experimental spectra, a large number of symmetric Lorentzians is used. However, not every peak needed for the fit has a physical meaning, e.g., the peak at 440 cm⁻¹. It could simply arise from the asymmetry of the p₂ peak. Why do the authors exclude asymmetric lineshapes in their fitting procedure?

Answer: We thank the reviewer for this input. In fact, as suggested by the Reviewer, the peak at 440 cm⁻¹ can arise from the asymmetry of the p₂ peak. Accordingly, we added the following comment in the revised version of the manuscript:

“The 440 cm⁻¹ peak was recently ascribed to the combination mode $A_{1g}+E_{2g}^2$ ²⁰. However, it can also be due to the asymmetric shape of p₂, since the vHs is not necessarily symmetric.”

In fact, as already discussed in our original manuscript (line 7, paragraph 2, page 4), the different contributions to the 2LA band are indeed asymmetric. This is clearly observed in our calculated results.

The fitting procedure of the experimental data with symmetric Lorentzians is only intended to provide a reliable estimate of the spectral position of the different contributions to the second-order Raman bands, as the central goal of this work is to achieve a quantitative comparison between the measured and the calculated spectra within the entire spectral range considered here. Asymmetrical peaks would also fit the 2LA band, but we would need to introduce more fitting parameters without physical meaning. The determination of the complex shape of each Raman feature would require a complete theoretical description of the Raman intensities, including electron-photon and electron-phonon coupling matrix elements.

In order to clarify this point, we added the following sentence in the revised version of the manuscript:

“The determination of the lineshape requires a complete theoretical description of Raman intensities, including electron-photon and electron-phonon coupling matrix elements.”

And the following in the same paragraph:

“Here we also use this procedure, but we stress that it is only intended to provide a reliable estimate of the spectral position of the different contributions to the second-order Raman bands. The central goal of this work is to achieve a quantitative comparison between the measured and the calculated spectra within the entire spectral range considered here while making use of a dense sampling in terms of laser energies.”

What are the different contributions to the error bars in Fig. 1(e),(f) and (h)?

Answer: We thank the reviewer for pointing this out. The error bars in Figs. 1(e), (f) and (h) represent the standard error from the fitting process for each laser energy. We have added the following sentence in the caption of Fig. 1:

“The error bars in Figs. 1(e), (f) and (h) represent the standard error from the fitting process.”

The authors claim that the p4 feature, which should be absent in a monolayer, has an almost negligible intensity in the single-layer sample. However, in Fig. 1(a) at an excitation energy of 2.11 eV, the p4 feature is of comparable intensity to the p1, p2, and p3 peaks. Hence, the interpretation of the p4 feature is not really sound.

Answer: We thank the reviewer for making this comment. First, we would like to emphasize that our conclusion is based on the comparison of the spectra of 1L and bulk MoS₂, and for different laser energies. In fact, the experimental intensity of p4 for 1L MoS₂ is weak, but not negligible as expected from the calculated data. This is possibly due to the strong electron-phonon coupling of p4. However, the main argument to support our interpretation of the p4 feature is the fact that p4 is weaker than p3 for 1L MoS₂ (direct gap semiconductor), and stronger than p3 for bulk MoS₂ (indirect gap semiconductor). In order to further confirm our conclusion, we performed additional measurements with samples consisting of 2L and 3L MoS₂, and these results are now shown in the Supplementary Fig. 3. Notice that in Suppl. Fig. 3, the shape of the 2LA band for the 1L MoS₂ sample is different from the shape of this band for 2L, 3L and bulk. For 1L MoS₂, the shoulder appears on the right side of the band, showing that p4 is weaker than p3, as expected for a direct gap system. On the other hand, for 2L, 3L and bulk, the shoulders appear on the left side, showing that p4 is stronger than p3, thus demonstrating that the transition between K points is possible, in agreement with the fact that they are indirect gap semiconductors. Therefore, the different behavior of p3 and p4 for 1L and for 2L, 3L and bulk MoS₂ proves that p4 has a real physical meaning, and it is not due to the asymmetry of p3.

In order to address this comment, we have added the following sentence in the revised version of the manuscript:

“Notice that different from the calculated results, the experimental intensity of p4 for 1L MoS₂ is weak but not negligible, this is possibly ascribed to the effect of a strong electron-phonon coupling of p4.”

However, the most relevant result is that p4 is more intense than p2 and p3 for 2L, 3L and bulk MoS2 (see Supplementary Fig. 3), and weaker for 1L MoS2.

In addition, and as pointed out by the Referee, the intensity of p4 at higher laser energies (e.g. near 2.11 eV) indeed increases for 1L MoS2, and becomes comparable with p1 (but still weaker than p2 and p3). Note that 2.11 eV is ~0.2 eV above the A exciton energy, which is exactly the misalignment energy between the conduction band state at Q and K (see Supplementary Fig. 1b). In this context, near 2.11 eV, the conduction band state at Q might become a resonant intermediate state (with a potentially large electron-phonon matrix element between K and Q). This explains the non-negligible intensity of p4 near 2.11 eV. However, p4 is still weaker than p2 and p3, whereas for bulk MoS2 at 2.11 eV it is stronger than p2 and p3, in agreement with our interpretation of p4.

In order to address this comment, we have added the following sentence in the Supporting Information:

“As shown in Fig. 1 of the main text, the intensity of p4 at 2.11 eV spectrum increases for 1L MoS2, and becomes comparable with p1 (but still weaker than p2 and p3). Note that 2.11 eV is ~0.2 eV above the A exciton energy, which is exactly the energy misalignment between the conduction band state at Q and K (see Supplementary Fig. 1b). That is, near 2.11 eV, the conduction band state at Q becomes a resonant intermediate state (with a potentially large electron-phonon matrix element between K and Q). This explains the non-negligible intensity of p4 near 2.11 eV.”

From the manuscript it doesn't become clear that for the measurements shown in Fig. 1(g) an ion bombarded sample is used. This should be mentioned more prominently.

Answer: We thank the reviewer for pointing this out. In order to emphasize that Fig. 1(g) refers to a bombarded sample, we have added the following sentence in Fig. 1g's caption:

“Defects were created through bombardment with Mn+ (see Experimental Methods section for details).”

Additionally, the Raman spectrum in the same spectral range as in Fig. 1(g) of a pristine flake should be shown.

Answer: We thank the reviewer for this suggestion. In order to address this comment, we have now added a new Supplementary Fig. 4 showing the Raman spectra of 1L and bulk (for 1.94, 2.04 and 2.11 eV), and the defective monolayer (for 1.92, 2.06 and 2.14 eV) in the spectral range of 200-500 cm^{-1} . Here, the pristine samples (1L and bulk) do not show any defective bands at around 230 cm^{-1} .

How does the ion bombardment influence the high-energy p1-p4 peaks? This would be an important information in order to get more insight into the intervalley scattering processes in the presence of defects.

Answer: We thank the reviewer for pointing this out. From the new Suppl. Fig. 4, we can observe that the 2LA band for the defective samples broadens and decreases in intensity, which is expected since the disruption of the pristine lattice decreases phonon lifetimes and introduces new defective Raman bands within this range.

The influence of the intervalley scattering effects described in the present work on a valley-polarisation of excitons should be discussed more extensively.

Answer: We thank the reviewer for this suggestion. In order to address this question, we have added a more detailed discussion regarding the valley-polarization of excitons, which appears immediately before the Discussion section, and it is attached here for convenience:

“Our conclusion that the LA(K) phonon dominates the intervalley scattering is consistent with previous reports of valley depolarization in TMDs³⁹⁻⁴¹. From the decay of the valley-polarization with increasing temperature, Zeng et al. extracted the energy of the phonon mode dominating intervalley scattering to be 240 cm⁻¹, consistent with the LA(K) phonon energy³⁹. However, this thermal destruction of valley polarization could also originate from the excitonic levels shifting as a function of temperature, thus detuning the laser excitation energy from the A exciton energy⁴⁰. To rule out this possibility, Kioseoglou et al. performed depolarization measurements as a function of increasing pump laser energy at T = 5 K, whereby the excess energy enables phonon-assisted intervalley scattering⁴¹. These authors found that the destruction of the valley polarization occurs beyond an excess energy of 60 meV (~ 480 cm⁻¹), which corresponds to two-LA(K) phonons, thus showing that intervalley scattering by LA phonons is responsible for depolarization⁴¹.”

Reviewer #2 (Remarks to the Author):

The paper by Carvalho et al. is in my opinion a very good piece of science. It is amazing how the authors provided a very detailed description of acoustic phonon dynamics in 2D MoS₂. The paper reports an extremely careful resonance Raman experiments by using many laser energies, a feature that is really needed for carrying out such experiments. The theoretical model employed at the state of the art in condensed matter physics fully support the experimental data and allowed the authors to further understanding the intervalley scattering by acoustic phonons using double resonance Raman spectroscopy. The detailed description of the defect induced modes is also a great advance that this paper brings to the community, especially the elucidation of the intervalley scattering by acoustic phonons and its consequence to valleytronics in these kind of materials. I strongly recommend the publication of the paper in Nature Communications because the results presented here are sound and will broadly impact the field of 2D materials, a hot topic today in materials science.

Answer: We thank the reviewer for the positive comments and recommendation of our manuscript for publication. Below we have addressed all the points raised by the Reviewer.

Before the final acceptance, the authors should comment some points.

Page 3. The frequency dependence of some modes is attributed to DRR mechanism. Is the DRR the unique way of having frequency dependence on laser excitation? The authors should clarify better this point.

Answer: This is a very interesting and fundamental question. It is well known that, in a normal second-order Raman process, the positions of the bands are the same when the spectrum is measured with different laser excitation energies. For a dispersive band, the intermediate step of the Raman process obtained with different laser energies selects phonons with different momenta. This selective process is only possible if this electron-phonon process is resonant. The most common process where the intermediate step is resonant is the double-resonance Raman mechanism, which has been used for several years in classical semiconductors [P. Yu and M. Cardona, Fundamentals

of Semiconductors: Physics and Materials Properties, Graduate Texts in Physics (Springer-Verlag, Berlin, Heidelberg, 2010)], and in carbon-based materials (e.g. graphene and nanotubes). However, any higher-order resonant process can give rise to laser energy dependence of some modes. We added a sentence in the revised version of the manuscript in order to discuss the latter:

“On the other hand, p_3 and p_4 exhibit a dispersive behavior, a signature of a DRR process (or higher-order resonant process) due to photons with different energies selecting electrons and phonons with different wave vectors in the BZ.”

Page 4. Is there any physical reason that would justify the linewidth fixed as a constraint for the Raman analysis?

Answer: Despite the dependence of the p_1 – p_4 peak positions on the laser energy, the peak linewidths are not expected to be significantly dependent on the laser energy, especially because we are investigating a narrow range of laser energies (1.85 to 2.18 eV). The weak dependence of the linewidth of the DRR bands has already been observed in graphene. This procedure was used to decrease the number of fitting parameters, in order to avoid fittings with different linewidths, without any physical meaning. We have added the following sentence to address this question:

“This procedure was adopted to decrease the number of fitting parameters, since the FWHM is not expected to depend significantly on the laser energy within this narrow range of energy (1.85–2.18 eV).”

Could the authors comment on the kind of defect will activate the disorder induced modes in MoS₂?

Answer: This is a very interesting and important question but, as far as we know, there is not yet experimental or theoretical study on the characterization of different types of defects by their (potentially different) Raman signatures. In principle, any kind of defect that breaks the translational symmetry can provide elastic scattering for momentum conservation in a DRR process. Even in the case of graphene, which has been studied for many decades, the distinction of different kinds of defects by the disorder induced Raman bands is not completely established. In our sample, ion bombardment could preferentially remove sulphur atoms, and we cannot exclude other vacancy-type such as metal atoms, as shown in atomic resolved STEM–ADF results [Nat. Commun. 6, 6293 (2015)]. Although this question is quite interesting, it is beyond the scope of the current work. A detailed study of irradiation conditions together with high-resolution transmission electron microscopy, resonant Raman and calculations would be required.

Is possible by using symmetry arguments predicts some edge dependence of disorder induced modes in MoS₂?

Answer: This is also an interesting point. It is known that the disorder-induced bands in graphene can distinguish different types of edges (armchair or zig-zag). However, determining edge types and edge reconstructions in further detail by simple symmetry arguments is not trivial, and requires alternative characterization methods such as electron microscopy. In the case of graphene, it involves the interplay between the polarization-dependence of both (1) the absorption (and emission) and the (2) electron-phonon interactions [PRL 93 247401, 2004]. In principle, a similar analysis can be performed to MoS₂. However, despite the fact that this is a very important issue, it is not in the scope of the present work. This issue might be a subject of future work.

How polarization depend measurements will help to further understand de intervalley scattering in MoS2?

Answer: We believe that it can provide further information about the anisotropy interaction of excited electrons with the acoustic phonons involved in the intervalley scattering. In the case of graphene, the spectra show strong polarization dependence for the double-resonance Raman intensities [Nano Letters 2008, 8 (12), pp 4270–4274]. It was showed that this strong polarization dependence is a direct consequence of inhomogeneous optical absorption and emission mediated by electron-phonon interactions involved in the second-order Stokes–Stokes Raman scattering process. For MoS₂, we find an anisotropy in optical transition matrix elements in the order of 15% (= 1–min/max) from preliminary first-principle calculations and, this is smaller than the 100% for the case of graphene. Although weak, this anisotropy may be utilized to probe electron-phonon interactions near the absorption edge. Such topics go beyond the scope of the current work and might be the subject of future work.

List of Changes:

During the revision of our manuscript, we have notice small typos that now are corrected and added some sentences all listed below:

1. Page 4:
 - a. 2nd paragraph, 2nd line, in the sentence "...Raman band is normally asymmetric and given by..." we have removed the words "...normally asymmetric and ...";
 - b. 2nd paragraph, 3rd line, the sentence "...across the BZ. In principle, a second-order Raman band cannot be fitted by a Lorentzian curve" was replaced by "...**across the whole BZ and, therefore, cannot be fitted by a sum of Lorentzian curves ...**";
 - c. 2nd paragraph, 4-5th line, we added "**The determination of the lineshape requires a complete theoretical description of Raman intensities, including electron-photon and electron-phonon coupling matrix elements.**";
 - d. 2nd paragraph 7-8th line, we added "**...by a sum of...**";
2. Page 4-5: the sentence "Here we also use this procedure, but carefully consider the physical origin of each feature while making use of a dense sampling in terms of laser energies" was replaced by "**Here we also use this procedure, but we stress that it is only intended to provide a reliable estimate of the spectral position of the different contributions to the second-order Raman bands. The central goal of this work is to achieve a quantitative comparison between the measured and the calculated spectra within the entire spectral range considered here while making use of a dense sampling in terms of laser energies.**";
3. Page5: we added "**This procedure was adopted to decrease the number of fitting parameters, since the FWHM is not expected to depend significantly on the laser energy within this narrow range of energy (1.85--2.18~eV).**"
4. Page 5:
 - a. 1st paragraph, 3rd line: "this broad band was fitted by three Lorentzian peaks" has been corrected to "this broad band was fitted by **four** Lorentzian peaks";
 - b. 1st paragraph, 3rd line: added "**...and, a peak at 440 cm-1 for 1L MoS2.**";
5. Page 8: "shown in see Fig. 2b,c,d." has been corrected to "**as shown** in Fig. 2b,c,d.";
6. Page 9:
 - a. 2nd paragraph, 3rd line: "represented in Fig. 3c,d." has been corrected to "**(Supplementary Fig. 1)**";
 - b. 2nd paragraph, 7th line: "in Figs. 3a,b" has been corrected to "in Figs. **3a-d**"; and we added the sentence: "**for 1L MoS2 and bulk, respectively,**"
7. Page 10-11:
 - a. Last paragraph page 10, 3rd line: The sentence "...the 2LA band was fitted by three peaks, p2; p3; and p4, as shown in Fig. 1a,b." has been corrected to "...the 2LA band was fitted by **four peaks, a peak around 440 cm-1 and the peaks p2, p3, and p4**, as shown in Fig. 1a,b.;

- b. 1st line, beginning of page 11: we added “(or higher-order resonant process)”;
8. Page 11:
- a. at the beginning of the page, we added “The 440 cm⁻¹ peak was recently ascribed to the combination mode $A_{1g}+E_{2g}^2$ ²⁰. However, it can also be due to the asymmetric shape of p2, since the vHs is not necessarily symmetric.”;
 - b. 1st paragraph, 9th line, the sentence “but is absent or very weak for 1L MoS2.”, the word “absent” has been removed.
 - c. At the end of the page, we added “Notice that different from the calculated results, the experimental intensity of p4 for 1L MoS2 is weak but not negligible, this is possibly ascribed to the effect of a strong electron-phonon coupling of p4. However, the most relevant result is that p4 is more intense than p2 and p3 for 2L, 3L and bulk MoS2 (see Supplementary Fig.3), and weaker for 1L MoS2.”;
9. Page 13: the sentence “in Fig. 2e,f” has been corrected to “in Fig. 2f”, and we added “for bulk MoS2”;
10. Page 14: we added “(see also Supplementary Fig. 4)”;
11. Page 17: we added the sentence “Our conclusion that the LA(K) phonon dominates the intervalley scattering is consistent with previous reports of valley depolarization in TMDs³⁹⁻⁴¹. From the decay of the valley-polarization with increasing temperature, Zeng *et al.* extracted the energy of the phonon mode dominating intervalley scattering to be 240 cm⁻¹, consistent with the LA(K) phonon energy³⁹. However, this thermal destruction of valley polarization could also originate from the excitonic levels shifting as a function of temperature, thus detuning the laser excitation energy from the A exciton energy⁴⁰. To rule out this possibility, Kioseoglou *et al.* performed depolarization measurements as a function of increasing pump laser energy at T = 5 K, whereby the excess energy enables phonon-assisted intervalley scattering⁴¹. These authors found that the destruction of the valley polarization occurs beyond an excess energy of 60 meV (\approx 480 cm⁻¹), which corresponds to two-LA(K) phonons, thus showing that intervalley scattering by LA phonons is responsible for depolarization⁴¹.”
12. In Figure 1’s caption two sentences were added:
- a. “Defects were created through bombardment with Mn+ (see Experimental Methods section for details).”; and,
 - b. “The error bars in Figs. 1(e), (f) and (h) represent the standard error from the fitting process.”;
13. In Figure 3a-d, the Raman frequencies (x-axis) has been rigidly shifted by within 1% to align with the experimental data, and we added the following sentence in the Theoretical methods: “Since the absolute Raman shifts are prone to slight (1%) over/under-estimates due to limitations of density functional theory, the calculated phonon frequencies were shifted by within 1% to allow better comparison with the respective experimental dispersions. This shift leaves the calculated *dispersion* in the Raman shift with laser energy (which reflects the influence of the phonon and electron band structures on the DRR process) unchanged.”
14. In Supplementary Information we have added two new figures, Supplementary Figs. 3 and 4, respectively.